# DATA-EFFICIENT ONLINE TRAINING FOR DIRECT ALIGNMENT IN LLMS

## ABSTRACT

In recent years, online Direct Alignment from Preferences (DAP) has emerged as a popular alternative for Reinforcement Learning from Human Feedback (RLHF) due to its training stability and simplicity. In online DAP, training relies on preference data, each composed of a question and a pair of large language model (LLM) responses. However, annotating preference data, i.e., generating responses for questions, is computationally expensive. To address this, we propose DOTA, a data selection framework that minimizes the cost of generating preference data, while still ensuring the quality of training. First, we propose a metric called Preference Perplexity (PFP) that enables us to design a low cost, gradient-based method to effectively estimate the contribution of each preference data point to model performance – critical to data selection. Second, rather than first generating responses for all candidate questions and then selecting preference data points by measuring their PFP, we design an iterative multi-armed bandit (MAB)-based strategy that only has to generate responses for a small subset of questions, without missing valuable data points. Experiments on UltraChat-200k and HH-RLHF across 13 downstream tasks demonstrate that DOTA reduces computation cost by a factor of three on LLaMA-3-8B, Qwen-3-4B, and Qwen-3-1.7B, without compromising training effectiveness. Code and data are available at https://anonymous.4open.science/r/DOTA-5CC5.

## 1 INTRODUCTION

A key challenge in developing safe and effective large language models (LLMs) lies in aligning their behavior with human preference through Reinforcement Learning from Human Feedback (RLHF) (Ouyang et al., 2022a; Bai et al., 2022a; Casper et al., 2023; Dai et al., 2023). The core of RLHF lies in the preference data (Stiennon et al., 2020; Poddar et al., 2024). A preference data point is in the form of $\{x, y^+, y^-\}$, where $x$ denotes the input question, $y^+$ represents the chosen LLM response, and $y^-$ represents the rejected response. An annotator is responsible for producing preference data (Wu et al., 2024; Yuan et al., 2023). Most recent research has shown that online DAP (Guo et al., 2024b; Qi et al., 2024b; Zhao et al., 2023; Garg et al., 2025), which leverages the target model to directly generate preference data to train the target model, improves the quality of model fine-tuning, because it ensures that the distribution of training data is consistent with the target model (Xiong et al., 2024; Guo et al., 2024a).

**Challenge.** Despite its advantages, online DAP suffers from expensive computational cost (Rafailov et al., 2023b; Lee et al., 2023; Zhong et al., 2024), because it has to (1) leverage the target model to generate responses for a large number of questions (denoted by a set $\mathcal{X}$), and then apply the reward model to annotate these responses; and (2) use those generated preference data points to train the target model. To reduce cost, several recent works have explored data selection in the context of online DAP. For example, Less-is-More (Deng et al., 2025) and SeRA (Ko et al., 2025) prioritize preference data points with clearer preference signals, i.e., data points with a large score gap between the chosen response $y^+$ and the rejected response $y^-$, where the scores are produced by the reward model. However, although these methods reduce the number of preference data points fed to the training phase, thus cutting down the training cost, they still have to generate responses for all candidate questions in $\mathcal{X}$ before measuring the score gap, which is costly.

**Our Proposal.** To the best of our knowledge, no work has targeted the problem of minimizing the cost of annotating reference data points in online DAP. To fill this gap, we propose a data selection framework `DOTA` that effectively and efficiently identifies a subset of questions whose corresponding reference data points potentially benefit the target model. `DOTA` features two key ideas: an effective metric to measure the value of a preference data point, called Preference Perplexity (`PFP`) and a multi-armed bandit based approach that leverages PFP to select valuable questions.

Preference Perplexity (`PFP`) solves the problem that the score gap produced by a *reward model*, which is used by existing methods to measure the value of a preference data point, does not necessarily reflect its potential benefit to the target model. We instead propose to estimate the value of a data point based on the *deviation* between the preference probability produced by the current target model and the expected preference probability produced by the reward model, because it exactly corresponds to the DAP loss and thus directly shows the impact of this data point on the current target model. However, computing the deviation in this way is computationally expensive as it requires two forward passes over both the target model and the reference model. We theoretically show that this deviation, in fact, can be effectively approximated based on the gradients of the DAP loss w.r.t. the target model. Leveraging this theoretical insight, we propose the PFP metric that enables `DOTA` to compute the deviation with only one single forward pass over the target model, offering an effective and lightweight solution to estimate the value of each reference data point.

Moreover, instead of selecting preference data points by first generating responses for all candidate questions and then computing their `PFP`, we propose an approach that only has to conduct this expensive process over a subset of carefully sampled questions. More specifically, `DOTA` first clusters questions in $\mathcal{X}$ such that data points within the same cluster exhibit similar deviation degrees. This allows `DOTA` to estimate the `PFP` of all questions within a cluster by sampling and annotating a small subset of questions. However, repeatedly sampling from a small number of high-`PFP` clusters tends to generate very similar preference data points. This redundancy causes diminishing returns during training. Therefore, we propose to balance the exploitation of high-`PFP` clusters with the exploration of the remaining clusters, by designing an iterative strategy based on multi-armed bandit (MAB). It computes the upper confidence bound (UCB) score for each cluster that penalizes the high `PFP` clusters if they are frequently sampled.

**Contributions.** The key contributions of this work include:

• We propose `DOTA`, the first data selection framework that minimizes the cost of annotating reference data points for online DAP methods, while ensuring the quality of training.
• We propose Preference Perplexity (`PFP`) that effectively estimates the potential value of each preference data point with low cost.
• We design an iterative MAB-based strategy that effectively leverages `PFP` to select a subset of valuable questions for annotation, so as to improve the performance of the target model.
• Extensive experiments on `HH-RLHF` and `UltraChat-200k` datasets as well as 13 popular downstream tasks demonstrate that `DOTA` significantly outperforms all baseline methods. Compared to training with a complete dataset, it saves $3\times$ computation resources with comparable accuracy. Compared to the state-of-the-art data selection methods, it saves $2\times$ computation resources, while improving accuracy by 2% (train/test on Llama-3-8B, Qwen-3-4B, and Qwen-3-1.7B models).

## 2 PRELIMINARY

Online DAP methods directly update the target model $\pi_\theta$ based on preference data points (i.e., $\{x, y^+, y^-\}$). The goal is to increase the probability of generating $y^+$, while decreasing that of $y^-$. This effectively aligns the target model $\pi_\theta$ with human preferences. Among all online DAP methods, online Direct Preference Optimization (DPO) (Rafailov et al., 2023a) has emerged as a representative and widely used approach (Guo et al., 2024b).

**Online DPO.** Adopting the Bradley–Terry model (Bradley & Terry), online DPO defines *preference probability* as a logistic function of the deviation between the scores of two responses, where the score denotes the probability of generating $y^+$ ($y^-$) given $x$ under the KL divergence constraint (i.e., $\beta \log \frac{\pi_\theta(y^+|x)}{\pi_{\text{ref}}(y^+|x)}$). Then it defines the training loss as the negative log of this preference probability. By minimizing this loss, online DPO maximizes the relative likelihood of generating $y^+$ over $y^-$,

steering $\pi_\theta$ towards human preference. Formally, the loss is defined as follows:

$$\mathcal{L}_{\text{DPO}} = -\log \sigma\left(\beta \log \frac{\pi_\theta(y^+ \mid x)}{\pi_{\text{ref}}(y^+ \mid x)} - \beta \log \frac{\pi_\theta(y^- \mid x)}{\pi_{\text{ref}}(y^- \mid x)}\right) \tag{1}$$

where $\pi_{\text{ref}}$ denotes the reference model (typically a frozen copy of the target model (Ouyang et al., 2022b)), which is used to constrain the distributional drift of $\pi_\theta$ using the KL divergence as a regularization term. During the training phase, the online DPO prompts the model $\pi_\theta$ with a question $x$ to generate a pair of responses $(y_1, y_2) \sim \pi_\theta(y \mid x)$. A reward model $r_\phi$, which serves as an annotator, scores them to produce an online preference data point $\{x, y^+, y^-\}$, which is then used to update the target model $\pi_\theta$ to reduce the deviation between $\pi_\theta$ and human preferences.

**Other online DAP Methods.** Similar to online DPO, many other online DAP methods have also been proposed to directly align LLMs with human preferences, such as online Identity Policy Optimization (online IPO) (Garg et al., 2025) and online Sequence Likelihood Calibration with Human Feedback (online SLiC) (Zhao et al., 2023).

Specifically, IPO replaces Bradley–Terry reward used in DPO with an objective that minimizes the squared log-probability gap between $y^+$ and $y^-$:

$$\mathcal{L}_{\text{IPO}} = \left[\left(\log \frac{\pi_\theta(y^+ \mid x)}{\pi_{\text{ref}}(y^+ \mid x)} - \log \frac{\pi_\theta(y^- \mid x)}{\pi_{\text{ref}}(y \mid x)}\right) - \frac{1}{2\beta}\right]^2 \tag{2}$$

SLiC directly maximizes the log-likelihood of $y^+$ while simultaneously minimizing that of $y^-$ without the need for any separate reward function, and a clip margin averts over-confident shifts, so preference alignment is achieved with supervised learning:

$$\mathcal{L}_{\text{SLiC}} = \max\left(0, 1 - \beta\left(\log \frac{\pi_\theta(y^+ \mid x)}{\pi_{\text{ref}}(y^+ \mid x)} - \log \frac{\pi_\theta(y^- \mid x)}{\pi_{\text{ref}}(y^- \mid x)}\right)\right) \tag{3}$$

## 3 THE DOTA METHOD

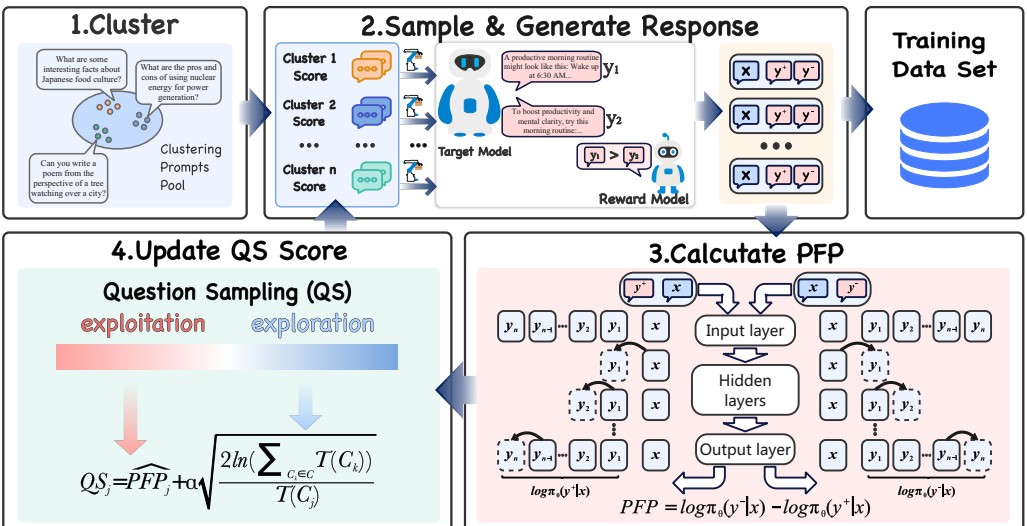

Figure 1: Overview of DOTA Framework

### 3.1 PROBLEM DEFINITION

In this paper, we address the problem of selecting data from a large pool of candidate questions to generate preference data points used by online DAP as training data to align LLMs with human preferences. Specifically, given a candidate question pool $\mathcal{X}$, we select a subset $\{x_i\}_{i=1}^n \subset \mathcal{X}$ to build a training set $\{x_i, y_i^+, y_i^-\}_{i=1}^n$, to fine-tune the target model $\pi_\theta$ with an online DAP method, where the objective is to minimize the loss of the updated model $\pi_\theta$.

## 3.2 THE OVERALL FRAMEWORK OF DOTA.

As shown in Figure 1, the DOTA framework can be divided into the following four steps. Firstly, DOTA *clusters all questions* in $\mathcal{X}$. After that, DOTA leverages multi-armed bandit to iteratively sample questions from each cluster. In each iteration, it selects some clusters and samples some questions from the selected cluster to generate preference data points and add to the training data set (Step-2). Then it computes the preference perplexity (PFP) scores for these data points that effectively represent their potential contribution to the target model (Step-3). Finally, taking into account the average PFP and the sampling frequency of each cluster, it computes the upper confidence bound (UCB) and selects the next set of clusters to sample (Step-4). The goal is to balance the exploitation of high-PFP clusters with the exploration of other rarely visited low-PFP clusters, ensuring the quality and diversity of the selected training set.

## 3.3 PREFERENCE PERPLEXITY (PFP)

The DAP loss is effective in measuring the potential value of each preference data point. This is because a large DAP loss indicates that the target model has not fitted this data point well. Therefore, selecting it as training data tends to better align the model with user preference. However, it is rather expensive to compute, as it requires two forward passes over both the target model and the reference model. To solve this problem, we propose Preference Perplexity (PFP) and theoretically show that it effectively approximates the DAP loss, while only requiring one forward pass over the target model.

For ease of presentation, we use online DPO as an example to illustrate DOTA. In the appendix, we discuss how DOTA can be applied to other DAP methods. Specifically, to elucidate how PFP reflects the update of the target model, we compute the gradient of DPO with respect to the model parameters $\theta$ as follows:

$$\nabla_\theta \mathcal{L}_{DPO} = -\beta\, \sigma\big(-\beta(\log \frac{\pi_\theta(y^+ \mid x)}{\pi_\theta(y^- \mid x)} - \log \frac{\pi_{\mathrm{ref}}(y^+ \mid x)}{\pi_{\mathrm{ref}}(y^- \mid x)})\big)\, \nabla_\theta \log \frac{\pi_\theta(y^+ \mid x)}{\pi_\theta(y^- \mid x)} \tag{4}$$

During the iterative training process of online DPO, at the beginning of each iteration, the target model is set the same as the reference model, and thus $\pi_{\mathrm{ref}} = \pi_\theta$. Then during training, the target model is constrained using the KL regularization, and thereby $\pi_{\mathrm{ref}} \approx \pi_\theta$ (Qi et al., 2024a; Chen et al., 2024; Schulman et al., 2017a; 2015), which is also validated in our experiment (Section 4.3).

Therefore, we can conduct Taylor expansion of the term $\beta(\log \frac{\pi_\theta(y^+|x)}{\pi_\theta(y^-|x)} - \log \frac{\pi_{\mathrm{ref}}(y^+|x)}{\pi_{\mathrm{ref}}(y^-|x)})$ and yields the following:

$$\beta\, \sigma\big(-\beta(\log \frac{\pi_\theta(y^+ \mid x)}{\pi_\theta(y^- \mid x)} - \log \frac{\pi_{\mathrm{ref}}(y^+ \mid x)}{\pi_{\mathrm{ref}}(y^- \mid x)})\big) = \frac{\beta}{2} - \frac{\beta^2}{4}\,(\log \frac{\pi_\theta(y^+ \mid x)}{\pi_\theta(y^- \mid x)} - \log \frac{\pi_{\mathrm{ref}}(y^+ \mid x)}{\pi_{\mathrm{ref}}(y^- \mid x)}) + O(\beta^3) \tag{5}$$

Simplifying the expressions above, we obtain

$$\nabla_\theta \mathcal{L}_{DPO} = \left(\tfrac{\beta}{2} + O(\beta)\right) \nabla_\theta(-\log \frac{\pi_\theta(y^+ \mid x)}{\pi_\theta(y^- \mid x)}) \tag{6}$$

where $\frac{\beta}{2} + O(\beta)$ can be considered a constant. From Equation 6, we observe that $\nabla_\theta \mathcal{L}_{\mathrm{DPO}}$ is determined by $\nabla_\theta\left(-\log \frac{\pi_\theta(y^+|x)}{\pi_\theta(y^-|x)}\right)$. Therefore, using $-\log \frac{\pi_\theta(y^+|x)}{\pi_\theta(y^-|x)}$ to approximate the DPO loss is still effective for DOTA to select preference data points, because ignoring the constant $\frac{\beta}{2} + O(\beta)$ will not alter the relative order of these data points, which determines their priorities. However, compared to the original DPO loss $\mathcal{L}_{\mathrm{DPO}}$ in Equation 1, which requires an additional forward pass through the reference model $\pi_{\mathrm{ref}}$, it is much more efficient to compute, especially suitable for the acquisition of large-scale online preference data.

Building on this, we propose Preference Perplexity (PFP) as an effective metric for data selection.

$$\mathrm{PFP}(\{x_i, y_i^+, y_i^-\}_{i=1}^N) = \sum_{i=1}^N (\log \pi_\theta(y^- \mid x) - \log \pi_\theta(y^+ \mid x)) \tag{7}$$

A larger PFP indicates a greater deviation of a preference data point, which in turn indicates a higher contribution to the target model, and vice versa.

In this way, to measure the contribution of a data point $\{x, y^+, y^-\}$, we only need to compute the probabilities of the target model predicting $y^+$ and $y^-$ given $x$ based on the next-token prediction (i.e., $\log \pi_\theta(y^+ \mid x)$ and $\log \pi_\theta(y^- \mid x)$). These probabilities have already been computed during the generation of preference data points using the target model $\pi_\theta$, and calculate the `PFP` based on the difference between these two probabilities (as shown in Figure 1).

**Bridging `PFP` with Perplexity (`PPL`) (Brown et al., 1992).** To further illustrate why the `PFP` in Equation 7 is effective in measuring the contribution of a preference data point to the target model, we rearrange it into Equation 9 and establish a connection with `PPL` (see Equation 8), which is broadly used in LLM instruction-tuning to measure the model confidence in next-token prediction when generating the response $y$ of a data point $(x, y)$:

$$\text{PPL}(y \mid x) = \exp\left(-\frac{1}{N} \sum_{j=1}^{N} \log p(y_j \mid x, y_1, \ldots, y_{j-1})\right) \tag{8}$$

$$\text{PFP}(x, y^+, y^-) = \log \pi_\theta(y^- \mid x) - \log \pi_\theta(y^+ \mid x) = |y^+| \cdot \log \text{PPL}(x, y^+) - |y^-| \cdot \log \text{PPL}(x, y^-) \tag{9}$$

In Equation 9 (`PFP`), $|y^+|$ and $|y^-|$ denote the lengths of responses $y^+$ and $y^-$, respectively. In Equation 8 (PPL), $N$ denotes the length of the response $y$ and $y_j$ represents the $j$-th token in the response $y$.

Equation 9 clearly shows the connection between `PFP` and `PPL`. We can immediately observe from it that a high `PFP` is likely to indicate a high $\text{PPL}(x, y^+)$ and a low $\text{PPL}(x, y^-)$. Consequently, the model is less confident in the chosen response $y^+$ and more confident in the rejected response $y^-$, indicating a large deviation from the human preference encoded in this reference data point. Therefore, these data points, if used in training, are valuable in aligning the target model with human preference, which is the goal of online DPO.

**`PFP` for other online DAP methods.** `PFP` equally works for these DAP methods (proof is provided in Appendix B, which is similar to that of DPO). Our experiments also confirm its effectiveness in other DAP methods.

### 3.4 THE DOTA ALGORITHM

In this section, we illustrate the details of the `DOTA` framework.

---

**Algorithm 1:** `DOTA` Algorithm

**Input:** Candidate questions pool $\mathcal{X}$, sampled data ratio $\gamma$, number of training iterations $M$, target model $\pi_\theta$.

**Output:** Well trained model $\pi_\theta^M$.

1  $\mathcal{C} = \texttt{Cluster}(\mathcal{X})$;
2  **for** $m = 1, \ldots, M$ **do**
3      Generated preference data $\mathcal{G} = \emptyset$;
4      **while** $|\mathcal{G}| < \gamma |\mathcal{X}|$ **do**
5          Select cluster $C_i$ with the highest QS score;
6          Sample questions $Q_i$ from $C_i$;
7          Generate preference data points $g_i$ from $Q_i$;
8          $\mathcal{G} = \mathcal{G} \cup g_i$;
9          $\widehat{\text{PFP}}_i = \mathcal{PFP}(\cup g_i)$, $T(C_i)\mathrel{+}=1$;
10         **for** $C_j$ *in* $\mathcal{C}$ **do**
11             $QS_j = \widehat{\text{PFP}}_j + \alpha\sqrt{\frac{2\ln \sum_{C_k \in \mathcal{C}} T(C_k)}{T(C_j)+1}}$;
12         **end**
13     **end**
14     $\pi_\theta^{m+1} = $ Train model $\pi_\theta^m$ with $\mathcal{G}$;
15 **end**
16 **return** $\pi_\theta^M$;

---

**QS Scoring with UCB.** Upper confidence bound (UCB) (Auer, 2002) is broadly adopted to balance 'exploration' and 'exploitation'. Specifically, clusters with a higher $\widehat{\text{PFP}}_i$ (see Section 3.3) have a higher chance to be selected, as it is more challenging for the model to correctly predict $\{x, y^+\}$ and $\{x, y^-\}$ associated with the question $x$. At the same time, clusters that are less frequently visited are prioritized as well, promoting the exploration of a more diverse set of questions. Based on the above insight, we define the Question Sampling (QS) score $QS_j$ of a cluster $C_j$ to effectively balance exploration (*i.e.*, data diversity) and exploitation (*i.e.*, data quality) as follows.

$$QS_j = \widehat{\text{PFP}}_j + \alpha\sqrt{\frac{2\ln \sum_{C_k \in \mathcal{C}} T(C_k)}{T(C_j)+1}} \tag{10}$$

where $T(C_j)$ denotes the frequency of questions sampled from cluster $C_j$, $\sum_{C_k \in \mathcal{C}} T(C_k)$ denotes the total number of samples from all clusters. $\alpha$ is set as $\frac{1}{\sum_{C_k \in \mathcal{C}} T(C_k)+1}$ (Hao et al., 2019), which assigns a higher weight to exploration in early stages and exploitation in later stages (Lines 10-12).

**Update the QS Score.** In each selection round, a subset of questions $Q_i$ is sampled from the selected cluster $C_i$ with the highest QS score (line 5); a set of preference data points $g_i$ (i.e., $\{x, y_n^+, y_n^-\}$) is then generated from questions in $Q_i$ (i.e., $x$). The PFP score of $C_i$ will be updated as follows.

$$\widehat{\text{PFP}_i} = \mathcal{PFP}(\cup g_i), \quad T(C_i) += 1 \tag{11}$$

where $\cup g_i$ denotes all the training data points from cluster $C_i$ generated at the beginning and $\mathcal{PFP}(\cdot)$ denotes the function of computing PFP score. Then we update the QS score of all clusters.

**Generated Pairs Collection.** As shown in Figure 1, in each selection round, we add the generated preference data points $g_i$ to the generated data pool $\mathcal{G}$ (Line 8 in Algorithm 1). Finally, in each training iteration $m$, we train model $\pi_\theta^m$ with the data points in $\mathcal{G}$. Then we apply the fine-tuned model $\pi_\theta^{m+1}$ to select the questions to be annotated, which will be used in the next training iteration.

# 4 EXPERIMENT

In this section, we fine-tune three base models on two benchmark datasets and conduct sufficient ablation studies to demonstrate the efficiency and effectiveness of DOTA.

## 4.1 EXPERIMENT SETUP

**Training Settings.** In both our warm-up SFT training phase and online DAP training phase, the batch size is set to 128. The learning rate is set to 5e-6 for both the Qwen-3 model and the Llama-3 model. For all DAP (i.e., DPO, IPO and SLiC) training, the loss parameter $\beta$ is set to 0.1. We adopt the Adam optimizer with hyperparameters $\beta_1 = 0.9$, $\beta_2 = 0.95$, and $\epsilon = 10^{-8}$. In each iteration, the model is trained for one epoch on the generated data points using eight A800 GPUs. Totally, we include three training iterations. During the warm-up stage, we randomly sample 5% of data from UltraFeedback to perform model warm-up training. For the clustering, we employ the BAAI/bge-large-en-v1.5 (Xiao et al., 2023) model to generate embeddings for candidate questions. Approximately 20k data points from the question candidate pool $\mathcal{X}$ are clustered using the $k$-means algorithm, with the number of clusters determined automatically (see Section 4.3). As for the reward model, we employ Skywork-Reward-V2-Llama-3.1-8B (Liu et al., 2025a), which holds the top position on the RewardBench2 (Malik et al., 2025) leaderboard[1]. The setting is consistent with that of Less is More (Deng et al., 2025).

**Dataset Preparation.** We use the popular alignment datasets UltraChat-200k (Tunstall et al., 2023) and HH-RLHF (Bai et al., 2022a) as our candidate datasets. Specifically, UltraChat-200k is a well-constructed and high-quality subset selected from UltraChat conversations. HH-RLHF contains human preference data collected by Anthropic, comprising two parts: helpful and harmless.

**Baselines.** We compare DOTA with several baselines: (1) SFT. The SFT model serves as the initialization for all online DAP methods. (2) Random. We design it following (Deng et al., 2025; Ko et al., 2025). In each iteration of online DAP, 30% of the questions are randomly sampled from the candidate question pool $\mathcal{X}$, and the generated preference data points are then used to train the target model, (3) Full. At each iteration, we train the target model using all preference data points generated from the candidate question pool $\mathcal{X}$. (4) SeRA (Ko et al., 2025). At each iteration, SeRA leverages implicit reward margins (IRM) to select the top 30% of preference data points from the candidate question pool $\mathcal{X}$ for training the target model. (5) Less is More (Deng et al., 2025). The Less Is More approach utilizes a dual-margin strategy that combines external and implicit reward scores to select the top 30% of generated online preference data points for model training. (6) Curry (Pattnaik et al., 2024). Curry sorts the generated preference data points in ascending order of the difference in reward scores evaluated by the reward model, allowing the target model to progressively learn from easier to more difficult examples. (7) DOTA-Top$k$. At each iteration, the target model generates preference data points for all questions $x$ in the candidate dataset. Then the top-30% online preference data points $\{x, y^+, y^-\}$ with the highest PFP scores are selected and used to train the target model. (8) DOTA-MAB uses PFP scores as rewards for the MAB to select 30% of the questions for data generation, which are then used to train the target model.

---

[1] RewardBench2 is a leading benchmark for evaluating the performance of reward models, covering diverse task domains with high evaluation difficulty and strong correlation to downstream performance

Table 1: Evaluation results of online DPO with 30% selection ratio. The highest scores are highlighted in bold and the the second- or third-highest scores are underlined. We run each experiment for three times and report the average.

| Models | General Tasks | | | | | | Mathematical | | Coding | | | RLHF Evalution | | | | |
|---|---|---|---|---|---|---|---|---|---|---|---|---|---|---|---|---|
| | EFLOPs | mmlu pro | drop | mmlu | agieval | korbench | gsm8k | math | humaneval | lcb | Average | Alpaca | Evol | Rreward | UltraFeed | Average |
| *Llama-3-8B* | | | | | | | | | | | | | | | | |
| SFT | - | 31.84 | 62.80 | 54.17 | 34.17 | 28.56 | 40.78 | 14.04 | 44.10 | 19.72 | 36.69 | - | - | - | - | - |
| Random | 5.880 | 32.95 | 66.18 | 59.08 | 36.63 | 30.80 | 44.23 | 15.50 | 47.05 | 21.74 | 39.25 | 57.68 | 56.48 | 56.84 | 57.85 | 57.21 |
| Curry | 5.912 | 33.21 | 66.07 | 59.68 | 36.15 | 30.44 | 44.54 | 15.28 | 47.31 | 21.53 | 39.36 | 57.87 | 57.08 | 57.04 | 57.57 | 57.39 |
| SeRA | 8.886 | 33.43 | 66.58 | 58.95 | 36.94 | 28.28 | 45.43 | 16.20 | 48.17 | 22.28 | 39.58 | 58.64 | 58.88 | 56.73 | 59.04 | 58.32 |
| Less is More | 11.289 | 35.12 | 67.41 | 61.05 | 37.93 | 32.40 | 47.29 | 16.62 | 51.39 | 24.80 | 41.56 | 60.24 | 59.64 | 59.87 | 60.14 | 59.97 |
| Full | 19.800 | 35.78 | 68.84 | 61.47 | 38.52 | 32.43 | 48.17 | 17.42 | 52.90 | 25.32 | 42.43 | 62.40 | 61.46 | 60.85 | 63.10 | 61.95 |
| DOTA(Top$k$) | 8.796 | 36.22 | 69.91 | 62.35 | 38.73 | 32.68 | 49.20 | 17.50 | 53.05 | 25.04 | 42.74 | 62.24 | 61.79 | 60.53 | 62.51 | 61.77 |
| DOTA(MAB) | 5.940 | 36.53 | 69.60 | 62.11 | 38.45 | 32.64 | 50.57 | 17.78 | 52.63 | 24.83 | 42.79 | 62.88 | 61.29 | 60.88 | 62.47 | 61.88 |
| *Qwen3-4B* | | | | | | | | | | | | | | | | |
| SFT | - | 44.75 | 79.58 | 61.47 | 38.25 | 49.84 | 28.98 | 19.56 | 58.51 | 17.31 | 44.28 | - | - | - | - | - |
| Random | 3.285 | 53.61 | 83.13 | 76.24 | 41.86 | 53.11 | 29.95 | 20.26 | 64.63 | 22.19 | 49.44 | 68.22 | 68.67 | 69.60 | 70.29 | 69.20 |
| Curry | 3.297 | 54.05 | 83.04 | 76.48 | 42.23 | 52.78 | 30.68 | 20.12 | 65.24 | 22.07 | 49.63 | 68.12 | 68.97 | 68.91 | 71.12 | 69.28 |
| SeRA | 4.428 | 54.61 | 83.21 | 76.83 | 42.67 | 52.80 | 30.33 | 20.42 | 64.54 | 24.33 | 49.97 | 69.12 | 68.54 | 69.34 | 71.83 | 69.71 |
| Less is More | 6.828 | 55.71 | 84.44 | 77.65 | 42.18 | 54.88 | 32.45 | 21.94 | 66.83 | 27.16 | 51.47 | 71.52 | 71.49 | 71.20 | 72.43 | 71.66 |
| Full | 10.950 | 56.28 | 85.35 | 78.02 | 44.98 | 55.04 | 32.84 | 23.02 | 69.80 | 30.74 | 52.90 | 73.66 | 72.24 | 70.40 | 74.43 | 72.68 |
| DOTA(Top$k$) | 5.448 | 56.97 | 85.18 | 78.22 | 44.69 | 55.28 | 33.21 | 22.92 | 70.73 | 28.14 | 52.82 | 72.55 | 73.62 | 71.40 | 74.33 | 72.98 |
| DOTA(MAB) | 3.315 | 56.53 | 84.96 | 77.93 | 44.55 | 55.60 | 32.95 | 23.12 | 68.68 | 29.10 | 52.60 | 72.73 | 73.62 | 71.00 | 74.66 | 73.00 |
| *Qwen3-1.7B* | | | | | | | | | | | | | | | | |
| SFT | - | 35.65 | 68.19 | 60.75 | 35.35 | 38.10 | 30.46 | 24.96 | 57.93 | 17.31 | 40.97 | - | - | - | - | - |
| Random | 1.755 | 38.92 | 71.29 | 63.37 | 37.15 | 40.24 | 32.95 | 26.26 | 57.54 | 18.28 | 42.89 | 70.18 | 67.46 | 70.02 | 69.16 | 69.21 |
| Curry | 1.773 | 39.17 | 71.15 | 63.55 | 37.04 | 40.84 | 33.21 | 25.87 | 58.37 | 18.07 | 43.03 | 70.42 | 66.66 | 70.49 | 69.45 | 69.25 |
| SeRA | 1.881 | 39.46 | 71.85 | 63.63 | 37.65 | 40.40 | 33.34 | 26.96 | 58.39 | 18.84 | 43.39 | 71.44 | 68.46 | 71.70 | 70.16 | 70.44 |
| Less is More | 3.981 | 41.77 | 72.33 | 64.68 | 38.15 | 42.12 | 35.71 | 27.58 | 61.23 | 19.66 | 44.80 | 72.42 | 68.05 | 72.40 | 71.28 | 71.04 |
| Full | 6.312 | 41.98 | 73.68 | 65.86 | 38.31 | 43.14 | 36.73 | 28.16 | 62.43 | 21.84 | 45.79 | 74.93 | 72.81 | 74.00 | 73.14 | 73.72 |
| DOTA(Top$k$) | 3.516 | 42.60 | 73.39 | 65.80 | 38.29 | 43.04 | 36.77 | 29.04 | 62.20 | 20.86 | 45.78 | 74.68 | 71.24 | 74.14 | 72.95 | 73.25 |
| DOTA(MAB) | 1.785 | 42.47 | 72.84 | 65.37 | 38.24 | 42.64 | 37.15 | 28.28 | 61.98 | 20.52 | 45.57 | 74.04 | 70.91 | 73.75 | 73.02 | 72.93 |

**Evaluation Metrics.** We assess the quality of the selected data by fine-tuning an LLM with these data points and evaluate its performance in the following two major aspects:

(1) Typical LLM Evaluation. We evaluate the capabilities of LLMs on well-known benchmark datasets across three major categories: (1) General Tasks: MMLU (Hendrycks et al., 2021), MMLU-PRO (Wang et al., 2024b), DROP (Dua et al., 2019), AGIEval (Zhong et al., 2023), KorBench (Ma et al., 2025); (2) Mathematical Tasks: GSM8K (Cobbe et al., 2021), MATH (Hendrycks et al.); (3) Coding Tasks: HumanEval (Chen et al., 2021), LCB (OpenCompass). We conduct all these evaluations with the OpenCompass (Contributors, 2023) framework.

(2) RLHF Evaluation. Following the setting of SeRA (Ko et al., 2024) and Less-is-More (Deng et al., 2025), we use GPT-4.1 as the evaluator to score the model-generated responses based on a given reference answer. Specifically, we report the win rate of the responses generated by the fine-tuned model and the responses generated by the initial SFT model (*e.g.*, Llama-3-8B-SFT). The win rate is evaluated with the following criteria: 1 point for a win, 0.5 points for a draw, and 0 points for a loss. All the prompts we used for evaluation are shown in Appendix F. For the datasets, we consider four popular test sets: AlpacaBench (Dubois et al., 2024), Evol-Instruct (Xu et al., 2025), RewardBench (Malik et al., 2025) and UltraFeedback (Cui et al., 2024).

(3) Efficiency Evaluation. To demonstrate the efficiency of different methods, we report the EFLOPs[2] (details in Appendix D) consumed by each method to quantify their GPU cost.

## 4.2 RESULTS

As shown in Table 1, DOTA outperforms all baseline methods on downstream tasks w.r.t. LLaMA-3-8B, Qwen-3-1.7B and Qwen-3-4B.

**Overall Performance.** As demonstrated in Table 1, our method DOTA(MAB) outperforms all baseline methods on all downstream tasks including both traditional LLM evaluation tasks and RLHF evaluation tasks. To be specific, on both tasks, DOTA(Top$k$) has an improvement of 3.5% and 4.5% respectively compared with Random on Llama-3-8B, confirming the effectiveness of PFP. DOTA outperforms Curry because it selects preference pairs where the chosen and rejected responses are farther apart only based on the reward model, without considering the impact on target model. Although SeRA and Less is More consider the performance of the target model, they do not perform well. This is because it selects preference samples $(x, y^+, y^-)$ with large differences in implicit reward (i.e., $y^+ \succ\succ y^-$). However, we note that such samples often contain simple or repetitive content, resulting in relatively small updates to the model parameters. DOTA(Top$k$) and DOTA(MAB) achieve similar performance, while DOTA-MAB is more efficient (saving 2.8 EFLOPs on Llama3-8B), as MAB promotes diversity in the question sampling process.

---

[2]FLOPs (Floating Point Operations) measure the total number of floating-point computations, serving as a standard metric for computational cost in LLMs. 1 EFLOPs corresponds to $10^{18}$ floating-point operations.

Table 2: Evaluation results of online IPO and SLiC with 30% selection ratio on Llama-3-8B. The highest scores are highlighted in bold and the the second- or third-highest scores are underlined. We run each experiment for three times and report the average.

| Other DAPs | General Tasks | | | | | | Mathematical | | Coding | | Average | RLHF Evalution | | | | Average |
|---|---|---|---|---|---|---|---|---|---|---|---|---|---|---|---|---|
| | EFLOPs | mmlu pro | drop | mmlu | agieval | korbench | gsm8k | math | humaneval | lcb | | Alpaca | Evol | Reward | UltraFeed | |
| **IPO** | | | | | | | | | | | | | | | | |
| Random | 3.273 | 45.24 | 80.64 | 63.61 | 39.94 | 50.00 | 37.12 | 21.97 | 59.37 | 12.66 | 45.62 | 70.46 | 70.84 | 71.22 | 72.55 | 71.27 |
| Curry | 3.285 | 45.29 | 80.36 | 63.50 | 40.23 | 50.03 | 37.43 | 22.07 | 59.76 | 12.89 | 45.73 | 70.80 | 71.12 | 71.13 | 72.17 | 71.31 |
| SeRA | 4.403 | 46.26 | 80.54 | 64.57 | 39.86 | 50.98 | 38.06 | 21.84 | 59.42 | 12.48 | 46.00 | 71.24 | 71.95 | 72.08 | 73.37 | 72.16 |
| Less is More | 6.798 | 46.37 | 81.83 | 65.77 | 40.58 | 52.23 | 40.33 | 23.54 | 61.87 | 14.09 | 47.18 | 72.44 | 72.68 | 72.65 | 74.58 | 73.09 |
| Full | 10.927 | 47.33 | 82.55 | 67.19 | 42.84 | 53.27 | 41.41 | 23.16 | 61.68 | 14.37 | 48.20 | 74.42 | 74.64 | 74.04 | 75.27 | 74.59 |
| DOTA(Top$k$) | 5.425 | 47.44 | 82.76 | 67.11 | 42.36 | 53.56 | 41.87 | 23.78 | 62.48 | 14.87 | 48.47 | 73.89 | 74.55 | 75.46 | 74.81 | 74.68 |
| DOTA(MAB) | 3.311 | 47.40 | 82.26 | 66.97 | 42.71 | 53.62 | 41.55 | 23.68 | 62.33 | 14.61 | 48.35 | 73.91 | 74.29 | 74.92 | 75.28 | 74.60 |
| **SLiC** | | | | | | | | | | | | | | | | |
| Random | 3.247 | 51.84 | 80.32 | 73.38 | 38.91 | 53.12 | 27.35 | 20.38 | 60.34 | 15.78 | 46.82 | 71.55 | 71.83 | 72.10 | 74.11 | 72.40 |
| Curry | 3.279 | 51.97 | 80.68 | 73.07 | 38.87 | 53.04 | 27.57 | 20.54 | 60.48 | 15.81 | 46.89 | 71.83 | 71.35 | 72.21 | 74.31 | 72.43 |
| SeRA | 4.387 | 51.89 | 80.30 | 74.02 | 38.98 | 53.44 | 27.23 | 21.13 | 61.49 | 15.38 | 47.10 | 72.48 | 72.35 | 73.08 | 74.98 | 73.22 |
| Less is More | 6.784 | 53.00 | 81.75 | 74.87 | 39.46 | 54.31 | 28.89 | 21.78 | 62.18 | 16.81 | 48.12 | 73.32 | 73.37 | 73.45 | 75.18 | 73.83 |
| Full | 10.897 | 53.39 | 82.54 | 75.15 | 40.12 | 55.04 | 30.66 | 22.75 | 62.93 | 17.48 | 48.90 | 75.16 | 75.51 | 75.00 | 76.02 | 75.42 |
| DOTA(Top$k$) | 5.418 | 53.60 | 82.67 | 75.09 | 40.07 | 54.74 | 31.01 | 22.87 | 62.95 | 17.93 | 48.99 | 74.32 | 75.40 | 74.10 | 76.54 | 75.09 |
| DOTA(MAB) | 3.305 | 53.28 | 82.41 | 75.04 | 40.04 | 54.88 | 29.87 | 22.45 | 63.01 | 17.78 | 48.75 | 75.03 | 76.08 | 74.39 | 76.68 | 75.41 |

In terms of FLOPs, except the methods (*i.e.*, `Random`, `Curry`) that use simple heuristics, `DOTA` consumes minimal computation resources because it only generates preference data points from a small subset of candidate questions, unlike `Topk`, `SeRA` and `Less is More`.

In addition, we also report the performance improvement of all methods across three training iterations, as illustrated in Figure 2. The x-axis denotes the number of iterations, while the y-axis represents the average score on downstream tasks. On all three models, our proposed `DOTA(MAB)` achieves consistent and notable gains over the three iterations, further demonstrating its robustness and effectiveness.

**Generalizability of `DOTA`.** To verify the general applicability of `DOTA`, we also conducted experiments on other widely used DAP methods such as IPO and SLiC. Specifically, we fine-tuned LLaMA-3-8B on 30% of the data selected by `DOTA`. As reported in Table 2, `DOTA` consistently outperforms all baselines on downstream tasks. This demonstrates that `DOTA`

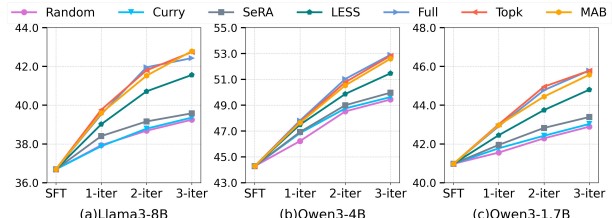

Figure 2: Number of training iterations on different models.

is applicable to different online DAP methods that use different training strategies.

Moreover, we also include `UltraChat-200k` (Tunstall et al., 2023) (`UltraChat`) as candidate question pool $\mathcal{X}$, from which 20%, 30% and 50% of the questions are selected to fine-tune Qwen-3-4B. As shown in Figure 3(a), only selecting 30% questions, `DOTA` achieves a performance comparable to using the entire dataset.

## 4.3 ABLATION STUDY

In this section, we conduct ablation studies *w.r.t.* the number of clusters, different clustering algorithms and the results are illustrated in Figure 3. Moreover, we show that $\pi_\theta$ and $\pi_{\text{ref}}$ exhibit similar confidence in generating the response $y$, thereby validating the approximation introduced in Section 3.3. We also demonstrate the effectiveness of MAB in `DOTA` through experiments under the `Topk-Clusters` setting, where questions are sampled from the top-$k$ clusters with the highest `PFP` scores to generate data points without iteratively exploration (see Appendix H for details).

**Number of Clusters.** We use the Elbow (Herdiana et al., 2025) method to identify the optimal cluster numbers for `HH-RLHF` and `UltraChat-200k` datasets. Figure 3(b) plots the accuracy of `DOTA` with different numbers of clusters. When the number of clusters is around 100, which is optimal for both

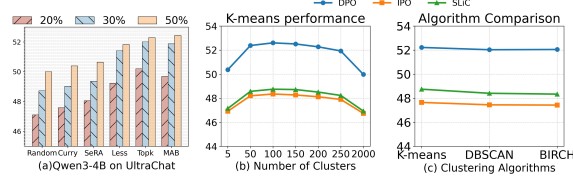

Figure 3: Qwen3-4B performance on UltraChat dataset and Ablation study of cluster numbers and algorithms.

datasets, the model consistently performs well. However, a very small number (*i.e.*, $k = 5$) leads to poor accuracy (4.2%, 3.0% and 3.3% lower accuracy for `DPO`, `IPO` and `SLiC`) due to the high variance of preference data points generated from the documents in each cluster. In this case, the sampled questions do not represent the cluster well. Similarly, when the number is too high (*i.e.*, $k = 2000$), there will be many clusters that generate similar preference data points. This jeopardizes

the diversity of the clusters that DOTA explores, leading to performance degradation – 5.3%, 3.3% and 3.7% lower accuracy.

**Clustering Algorithms.** We evaluate the performance of DOTA when using other typical clustering algorithms including BIRCH (Zhang et al., 1996a) and DBSCAN (Deng, 2020). The details of selecting optimal clustering parameters can be found in the Appendix E. As illustrated in Figure 3(c), DOTA is robust to clustering algorithms.

**Relation between $\pi_\theta$ and $\pi_{\text{ref}}$.** Figure 4 shows the results on the HHRLHF and UltraChat datasets using the Llama-3 and Qwen-3 models. Each point in the figure corresponds to a single preference data point, where the x-axis represents the deviation of preference data points on the reference model, measured as $\log \frac{\pi_{\text{ref}}(y^+|x)}{\pi_{\text{ref}}(y^-|x)}$ and y-axis denotes the

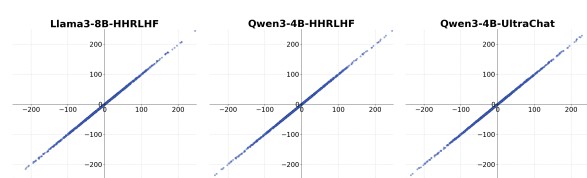

Figure 4: The similarity between $\pi_\theta$ and $\pi_{\text{ref}}$ on multiple models and datasets.

deviation of data points on the target model, i.e., $\log \frac{\pi_\theta(y^+|x)}{\pi_\theta(y^-|x)}$. The scatter points in Figure 4 exhibit a clear linear trend, indicating that the deviations of the target and reference models are nearly identical across all data points, thereby supporting the approximation introduced in Section 3.3.

## 5 RELATED WORK

**Alignment with preference data and DAP.** In preference-based alignment, conventional offline methods in RLHF typically first train a reward model on preference data $(x, y^+, y^-)$ (Bai et al., 2022b; Wang et al., 2024a), and then optimize the target model via reinforcement learning (e.g., PPO (Schulman et al., 2017b)). These two-stage methods are often unstable and vulnerable to reward hacking (Zhong et al., 2024; Peng et al., 2023). In contrast, Direct Alignment from Preferences (DAP) methods leverage preference pairs as direct supervision in training procedure (Guo et al., 2024b), surpassing explicit reward modeling. Representative methods include DPO (Rafailov et al., 2023a), which analytically derives the optimal policy and minimizes a preference-classification loss with a KL regularizer to maintain stability; IPO (Garg et al., 2025), which frames the problem through an identity and preference-optimization lens that sidesteps Bradley–Terry assumptions and directly optimizes a margin-based objective; SLiC (Zhao et al., 2023), which constructs a calibrated, comparison-driven supervision signal that jointly upweights preferred and suppresses dispreferred responses. Online preference alignment tackles the off-policy drift and narrow coverage of offline pipelines by generating responses on policy during training and immediately collecting preference signals (Song et al., 2024; Tajwar et al., 2024; Guo et al., 2024a).

**Data Selection.** Data selection methods aim to identify which candidate data points should be included in the training dataset (Albalak et al., 2024; Qin et al., 2024), as the quality of data points can vary significantly. Data selection w.r.t. both the pretraining and instruction tuning stages of LLMs has been extensively explored, where the research primarily focuses on improving data quality (Liu et al., 2025b), enhancing diversity (Zhang et al., 2024), and achieving distribution alignment (Liu et al., 2023). However, existing data selection methods in online DAP still require generating responses for all candidate questions before evaluating their values, which is computationally expensive.

## 6 CONCLUSION

We presented DOTA, a framework for data-efficient online DAP that addresses the high computational cost of sampling and labeling preference data points. By introducing Preference Perplexity (PFP) to quantify the value of the preference data points to the target model, DOTA design a Multi-armed Bandit-based strategy that effectively produces high-value reference data points, while only having to annotate a small number of sample questions. Experiments on multiple datasets and LLM backbones show that DOTA reduces computation by up to 3× while improving the alignment of the target model with human preference, offering a practical solution for scalable preference-based training.

## ETHICS STATEMENT

This work adheres to the ICLR Code of Ethics. In this study, no human subjects or animal experimentation was involved. All datasets used, were sourced in compliance with relevant usage guidelines, ensuring no violation of privacy. We have taken care to avoid any biases or discriminatory outcomes in our research process. No personally identifiable information was used, and no experiments were conducted that could raise privacy or security concerns. We are committed to maintaining transparency and integrity throughout the research process.

## REPRODUCIBILITY STATEMENT

We have made every effort to ensure that the results reported in this paper are fully reproducible. Our experiments are conducted with clearly specified datasets, model architectures, and hyperparameters. All data preprocessing steps, training procedures, and evaluation metrics are explicitly described in the main text and supplementary materials. The codebase used for all experiments will be released publicly upon publication, including scripts for training, evaluation, and data preparation. Additionally, we provide pre-trained models and detailed instructions for reproducing all reported results. We also include sufficient ablation studies and analysis to allow independent verification of our claims.

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

## A    THE USE OF LARGE LANGUAGE MODELS (LLMS)

In the preparation of this work, we used LLMs as auxiliary tools in a limited capacity. Specifically, LLMs assisted in drafting portions of the code and in refining the wording of certain sentences for clarity and readability. All technical content, including the design of algorithms, experimental methodology, analysis, and interpretations, was independently developed by the authors. The use of LLMs was confined to language refinement and coding suggestions, and did not influence the scientific contributions or results reported in this paper.

## B    PFP FOR OTHER ONLINE DAP METHODS

**IPO** Specifically, following the approach of online DPO, we first compute the derivative of the `IPO` loss.

$$\nabla_\theta \mathcal{L}_{\text{IPO}} = -\frac{2}{\beta}\left(\frac{1}{2} - \beta(\log\frac{\pi_\theta(y^+ \mid x)}{\pi_\theta(y^- \mid x)} - \log\frac{\pi_{\text{ref}}(y^+ \mid x)}{\pi_{\text{ref}}(y^- \mid x)})\right)\nabla_\theta\frac{\log\pi_\theta(y^+ \mid x)}{\log\pi_\theta(y^- \mid x)} \tag{12}$$

As discussed above, when $\beta\left(\log\frac{\pi_\theta(y^+|x)}{\pi_\theta(y^-|x)} - \log\frac{\pi_{\text{ref}}(y^+|x)}{\pi_{\text{ref}}(y^-|x)}\right)$ is small the coefficient $\frac{1}{2} - \beta\left(\log\frac{\pi_\theta(y^+|x)}{\pi_\theta(y^-|x)} - \log\frac{\pi_{\text{ref}}(y^+|x)}{\pi_{\text{ref}}(y^-|x)}\right)$ can be approximated as a near-constant. In this regime, the effective gradient magnitude is primarily governed by the target model margin term $-\log\frac{\pi_\theta(y^+|x)}{\pi_\theta(y^-|x)}$. This implies that preference data points with larger values of $-\log\frac{\pi_\theta(y^+|x)}{\pi_\theta(y^-|x)}$ induce stronger corrective updates, whereas preference data points with smaller values lead to vanishing gradients and exert only limited influence on training. Consequently, prioritizing preference data points with arger values of $-\log\frac{\pi_\theta(y^+|x)}{\pi_\theta(y^-|x)}$ naturally corresponds to a hard-sample prioritization strategy, thereby enhancing the overall effectiveness of gradient-based optimization.

**SLiC** Similarly, for the `SLiC` method, we also take the derivative of its loss:

$$\nabla_\theta\mathcal{L}_{\text{SLiC}} = -\beta\,\mathbf{1}_{\left\{\beta\left(\log\frac{\pi_\theta(y^+|x)}{\pi_\theta(y^-|x)} - \log\frac{\pi_{\text{ref}}(y^+|x)}{\pi_{\text{ref}}(y^-|x)}\right)<1\right\}}\left[\nabla_\theta\frac{\log\pi_\theta(y^+ \mid x)}{\log\pi_\theta(y^- \mid x)}\right] \tag{13}$$

Here, $\mathbf{1}_{\{\cdot\}}$ serves as an indicator function, ensuring that only preference data points with small $\beta\left(\log\frac{\pi_\theta(y^+|x)}{\pi_\theta(y^-|x)} - \log\frac{\pi_{\text{ref}}(y^+|x)}{\pi_{\text{ref}}(y^-|x)}\right)$ contribute to the update. Within this region, the magnitude of the gradient is essentially controlled by the policy margin term $-\log\frac{\pi_\theta(y^+|x)}{\pi_\theta(y^-|x)}$, so prioritizing samples with larger $-\log\frac{\pi_\theta(y^+|x)}{\pi_\theta(y^-|x)}$ naturally corresponds to a hard sample strategy that improves the efficiency of optimization.

## C    DATASET

We use the popular alignment datasets `UltraFeedback` (Cui et al., 2023), `UltraChat-200k` (Tunstall et al., 2023) and `HH-RLHF` (Bai et al., 2022a) as our candidate dataset. Specifically, `UltraFeedback` is a preference dataset collected from diverse sources, which is typically used for RLHF (Deng et al., 2025; Ko et al., 2025). `UltraChat-200k` is a well-constructed and high-quality subset selected from UltraChat conversations. `HH-RLHF` contains human preference data collected by Anthropic, comprising two parts: helpful and harmless. Initially, these models are fine-tuned on `UltraFeedback` followed by an alignment with online DAPs and prompts from `UltraChat-200k` and `HH-RLHF`.

## D    FLOPS CALCULATION

FLOPs is the number of floating point operations performed by GPUs. Many state-of-the-art methods (Yu et al., 2024) use it to measure the consumption of GPU computing resources. In our experiments, FLOPs is collected directly in the data selection process using the Python code:

```
import torch
import torch.nn as nn
from torch.profiler import profile, ProfilerActivity

model = nn.Linear(1024, 512).cuda()
input_data = torch.randn(128, 1024).cuda()
with                      profile(activities=[ProfilerActivity.CPU,
ProfilerActivity.CUDA],
    with_flops=True) as prof:
        model(input_data)
print (prof.key_averages().table(sort_by="flops", row_limit=10))
```

## E   ABLATION STUDY OF CLUSTERING NUMBERS AND ALGORITHM

[*Metric and Criteria.*] We use the metric Within-Cluster Sum of Squares (WCSS) to select the best cluster number using the well-known Elbow (Syakur et al., 2018) algorithm. WCSS is the sum of squared distances between each data instance and its cluster center, i.e., WCSS=$\sum_{i=1}^{k} \sum_{x \in C_i} \|x - \mu_i\|$. At a high level, the criteria should be that within each cluster, data instances are close to each other, based on which it is better for different cluster centers to be far away from each other. Based on the criteria, the Elbow algorithm leverages the WCSS as a measurement to iteratively select an appropriate cluster number, as follows.

[*Specific hypermarameter selection strategy.*] To be specific, Elbow begins with a small $k$, and with $k$ increasing, WCSS first decreases rapidly and then slows down. Then, we identify the "elbow point" where the decreasing rate becomes slow as the best $k$. Thus, within each cluster, data points are sufficiently close to one another. Furthermore, given that $k$ remains modest, different cluster centers tend to maintain a distance from each other.

*Clustering algorithms.* In terms of the clustering algorithms, we also added experiments to show that EQUAL is not sensitive to clustering algorithms mainly because different algorithms have their own strategies to select appropriate parameters, which follows the criteria mentioned above.

Specifically, we evaluate the performance of several typical clustering methods including BIRCH (Zhang et al., 1996b) and DBSCAN (Ester et al., 1996). Considering that the clustering results are easily affected by the parameters of clustering algorithms, we use different methods to select proper parameters. For DBSCAN, there are 2 key parameters: (1) $eps$(the radius of a neighborhood w.r.t. some data points) and (2) $minPts$ (a data point is considered as a core point if at least $minPts$ data points are within $eps$ of it). They can be set using the method in (Schubert et al., 2017). For BIRCH (Zhang et al., 1996b), we can use the Elbow (Syakur et al., 2018) algorithm or Sihouette score (Shahapure & Nicholas, 2020) to determine the appropriate number of components.

## F   PROMPT

```
<|im_start|>system
You are a helpful instruction-following assistant.
<|im_end|>
<|im_start|>user
Select the output (a) or (b) that best matches the given instruction.
    Choose your preferred output, which can be subjective. Your answer
    should ONLY contain: Output (a) or Output (b). Here's an example:

# Example:
## Instruction:
Give a description of the following job: "ophthalmologist"

## Output (a):
An ophthalmologist is a medical doctor who specializes in the diagnosis
    and treatment of eye diseases and conditions.

## Output (b):
```

```
An ophthalmologist is a medical doctor who pokes and prods at your eyes
    while asking you to read letters from a chart.

## Which is best, Output (a) or Output (b)?
Output (a)

Here the answer is Output (a) because it provides a comprehensive and
    accurate description of the job of an ophthalmologist. In contrast,
    output (b) is more of a joke.

# Task:
Now is the real task, do not explain your answer, just say Output (a) or
    Output (b).

## Instruction:
{instruction}

## Output (a):
{output_1}

## Output (b):
{output_2}

## Which is best, Output (a) or Output (b)?
<|im_end|>
```

Listing 1: Experiment Prompt

## G    DATA SELECTION RATIO.

Figure 5 presents a comparison between DOTA and other baselines using various data generation ratios (i.e., 20%, 30% and 50%). Across all proportions, the performance of DOTA surpasses that of the other baseline methods. Interestingly, we

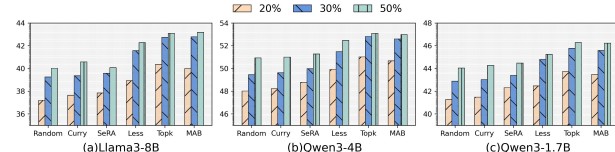

Figure 5: Ablation study of different data selection ratio

find that for all models, generating only 30% of the data points in DOTA yields better results than using the full dataset $\mathcal{X}$. This demonstrates the effectiveness of DOTA. This is because not all data points generated from the questions in $\mathcal{X}$ are highly valuable for the model's training. The detailed results are shown as follows.

Table 3: Comprehensive Evaluation of Three Models Across Diverse Downstream Tasks, Including General, Mathematical, and Coding Benchmarks. Evaluation results of various data selection methods at a 50% selection ratio. The highest scores are highlighted in bold, the second-highest scores are underlined, and the methods proposed in this work are marked in light yellow.

| Models | time | General Tasks | | | | | Mathematical | | Coding | | Average |
|---|---|---|---|---|---|---|---|---|---|---|---|
| | EFLOPs | mmlu | mmlu pro | drop | agieval | korbench | gsm8k | math | humaneval | lcb | Average |
| *Llama3-8B* | | | | | | | | | | | |
| random | 9.900 | 60.76 | 36.26 | 67.61 | 39.38 | 30.56 | 47.92 | 16.20 | 48.78 | 21.58 | 41.01 |
| SeRA | 12.030 | 59.28 | 35.33 | 64.70 | 37.63 | 29.00 | 39.98 | 15.20 | 48.78 | 22.52 | 39.06 |
| Less is More | 13.050 | 61.31 | 37.12 | 67.96 | 39.85 | 30.80 | 46.55 | 16.36 | 49.39 | 23.78 | 41.45 |
| Curry | 9.930 | 58.87 | 36.13 | 66.07 | 38.68 | 31.36 | 51.18 | 15.62 | 48.78 | 18.52 | 40.58 |
| Dota(Topk) | 11.940 | 62.45 | 37.74 | 70.05 | 40.27 | 31.20 | 50.42 | 17.18 | 50.00 | 22.12 | 42.38 |
| Dota(MAB) | 9.960 | 62.11 | 37.53 | 69.60 | 39.45 | 33.64 | 50.57 | 16.78 | 52.83 | 23.33 | 42.89 |
| *Qwen3-4B* | | | | | | | | | | | |
| random | 5.475 | 77.50 | 55.90 | 84.07 | 41.82 | 53.96 | 29.34 | 22.00 | 65.85 | 26.50 | 50.77 |
| SeRA | 6.000 | 76.45 | 54.14 | 83.31 | 41.52 | 54.15 | 29.76 | 16.93 | 58.23 | 29.05 | 49.28 |
| Less is More | 8.100 | 77.51 | 55.99 | 84.04 | 42.39 | 54.08 | 31.08 | 21.32 | 68.29 | 29.20 | 51.54 |
| Curry | 5.490 | 77.48 | 56.00 | 83.83 | 42.30 | 55.12 | 30.11 | 21.14 | 66.46 | 29.87 | 51.48 |
| Dota(Topk) | 7.020 | 77.61 | 56.80 | 84.66 | 43.27 | 55.04 | 32.90 | 23.40 | 70.73 | 31.06 | 52.83 |
| Dota(MAB) | 5.490 | 78.00 | 56.52 | 84.43 | 42.52 | 55.20 | 34.72 | 22.40 | 67.07 | 30.03 | 52.32 |
| *Qwen3-1.7B* | | | | | | | | | | | |
| random | 2.925 | 65.54 | 42.20 | 71.18 | 37.91 | 42.56 | 31.69 | 26.44 | 53.66 | 21.62 | 43.64 |
| SeRA | 2.547 | 64.49 | 40.88 | 71.42 | 37.38 | 42.56 | 34.12 | 27.88 | 53.05 | 19.52 | 43.48 |
| Less is More | 4.647 | 66.14 | 42.21 | 71.61 | 37.89 | 43.24 | 32.31 | 27.42 | 56.49 | 21.88 | 44.84 |
| Curry | 2.940 | 65.76 | 42.19 | 71.04 | 37.29 | 42.64 | 32.52 | 26.96 | 56.71 | 21.45 | 44.06 |
| Dota(Topk) | 4.182 | 66.93 | 42.33 | 72.59 | 38.00 | 44.40 | 34.04 | 27.74 | 60.37 | 21.92 | 45.37 |
| Dota(MAB) | 2.940 | 66.72 | 42.54 | 71.76 | 38.39 | 43.28 | 34.12 | 27.94 | 61.59 | 22.45 | 45.42 |

Table 4: Comprehensive Evaluation of Three Models Across Diverse Downstream Tasks, Including General, Mathematical, and Coding Benchmarks. Evaluation results of various data selection methods at a 30% selection ratio. The highest scores are highlighted in bold, the second-highest scores are underlined, and the methods proposed in this work are marked in light yellow.

| Models(20%) | time | General Tasks | | | | | Mathematical | | Coding | | Average |
|---|---|---|---|---|---|---|---|---|---|---|---|
| | EFLOPs | mmlu | mmlu pro | drop | agieval | korbench | gsm8k | math | humaneval | lcb | Average |
| *Llama3-8B* | | | | | | | | | | | |
| random | 3.960 | 59.41 | 33.28 | 62.45 | 37.33 | 29.44 | 45.49 | 16.42 | 48.78 | 20.22 | 39.20 |
| SeRA | 7.314 | 58.34 | 30.13 | 55.33 | 36.55 | 28.60 | 41.55 | 13.28 | 48.78 | 17.98 | 36.84 |
| Less is More | 9.954 | 59.02 | 33.38 | 57.29 | 37.47 | 29.52 | 45.98 | 14.24 | 53.66 | 19.79 | 38.93 |
| Curry | 3.990 | 58.83 | 33.19 | 66.07 | 37.32 | 29.28 | 45.79 | 15.84 | 50.00 | 20.50 | 39.65 |
| Dota(Topk) | 7.224 | 59.74 | 33.66 | 64.30 | 38.13 | 30.24 | 46.32 | 16.42 | 53.67 | 20.87 | 40.37 |
| Dota(MAB) | 3.990 | 59.83 | 33.90 | 66.08 | 37.43 | 29.84 | 45.81 | 16.48 | 50.00 | 20.54 | 39.99 |
| *Qwen3-4B* | | | | | | | | | | | |
| random | 2.190 | 77.52 | 55.95 | 83.63 | 42.69 | 54.48 | 31.99 | 22.94 | 63.41 | 24.86 | 50.84 |
| SeRA | 3.642 | 76.23 | 54.57 | 83.16 | 18.10 | 54.08 | 32.45 | 20.04 | 57.32 | 22.56 | 46.50 |
| Less is More | 5.243 | 77.58 | 56.39 | 84.08 | 42.18 | 53.60 | 30.63 | 20.30 | 64.63 | 28.60 | 50.90 |
| Curry | 2.205 | 77.35 | 56.07 | 83.39 | 42.77 | 54.40 | 32.75 | 22.88 | 65.85 | 27.07 | 51.39 |
| Dota(Topk) | 4.650 | 77.95 | 56.58 | 84.22 | 42.85 | 55.20 | 33.89 | 23.16 | 67.68 | 26.88 | 52.02 |
| Dota(MAB) | 2.205 | 77.73 | 56.16 | 83.96 | 42.73 | 54.56 | 33.43 | 22.76 | 68.90 | 27.03 | 51.92 |
| *Qwen3-1.7B* | | | | | | | | | | | |
| random | 1.170 | 64.70 | 41.42 | 71.12 | 38.11 | 42.32 | 35.71 | 27.96 | 58.54 | 20.04 | 44.46 |
| SeRA | 1.185 | 62.50 | 38.65 | 71.39 | 36.82 | 43.20 | 25.40 | 24.96 | 47.56 | 21.46 | 41.32 |
| Less is More | 3.648 | 65.25 | 41.67 | 70.93 | 37.42 | 41.52 | 35.33 | 28.58 | 59.76 | 19.68 | 44.46 |
| Curry | 1.185 | 64.72 | 41.09 | 71.29 | 38.53 | 42.16 | 35.86 | 28.18 | 56.10 | 20.66 | 44.29 |
| Dota(Topk) | 3.183 | 64.84 | 41.71 | 71.68 | 38.31 | 42.56 | 36.24 | 28.14 | 59.15 | 20.26 | 44.74 |
| Dota(MAB) | 1.185 | 65.64 | 41.50 | 71.38 | 38.62 | 42.57 | 37.00 | 28.74 | 61.59 | 20.51 | 45.28 |

Table 5: Evaluation results of online DPO with 30% selection ratio. We run each experiment three times and report the average.

| Models | EFLOPs | General Tasks | | | | | Mathematical | | Coding | | Average |
|---|---|---|---|---|---|---|---|---|---|---|---|
| | | mmlu pro | drop | mmlu | agieval | korbench | gsm8k | math | humaneval | lcb | |
| *Qwen3-4B* | | | | | | | | | | | |
| Random | 3.285 | 53.61 | 83.13 | 76.24 | 41.86 | 53.11 | 29.95 | 20.26 | 64.63 | 22.19 | 49.44 |
| Top$k$-Cluster | 5.448 | 56.97 | 85.18 | 78.22 | 44.69 | 55.28 | 33.21 | 22.92 | 70.73 | 28.14 | 50.70 |
| DOTA(MAB) | 3.315 | 56.53 | 84.96 | 77.93 | 44.55 | 55.60 | 32.95 | 23.12 | 68.68 | 29.10 | 52.60 |

# H  EFFECTIVENESS OF MAB

This section assesses the effectiveness of the MAB approach in `DOTA` for question selection in generating preference data points, as opposed to the simpler strategy of using the top-$k$ clusters with the highest `PFP` scores for the same purpose. To be specific, we randomly select an equivalent number of data points from the top 30 clusters with the highest `PFP` scores. Table 5 illustrates the trade-off between data quality and diversity: preference data points generated from high-`PFP` clusters within the `Topk-clusters` do not necessarily improve model performance on downstream evaluation tasks, as they often lack sufficient diversity. Hence, the multi-armed bandit method can more effectively capture the trade-off between quality and diversity across clusters, resulting in superior performance, as opposed to merely choosing the top-$k$ clusters.

