# OpenReview forum: "Data-efficient Online Training for Direct Alignment in LLMs"
_ICLR.cc/2026/Conference — ICLR 2026 Conference Withdrawn Submission_

### Official Review · Reviewer_2h1M · 2025-10-30

**Soundness:** 3
**Presentation:** 4
**Contribution:** 2
**Rating:** 4
**Confidence:** 4

**Summary:**

This paper introduces DOTA, a data selection framework designed to reduce the computational cost of generating preference data for Direct Alignment from Preferences (DAP). The authors propose a new metric, Preference Perplexity (PFP), which estimates the contribution of each preference pair to model performance, enabling more efficient data selection. They further develop an iterative multi-armed bandit (MAB) strategy that selectively generates responses for only a subset of questions instead of all candidates, thereby saving computation while preserving data quality. Experiments demonstrate that DOTA reduces computation cost without degrading a lot alignment or downstream performance.

**Strengths:**

- This paper proposes a data selection framework that effectively minimizes the cost of annotation, while maintaining training quality. The proposed selection algorithm is also computationally efficient, requiring only a single forward pass.

- The proposed DOTA framework achieves higher efficiency without sacrificing performance, as it ensures sufficient data diversity during selection.

- The writing is clear and well-organized, making the paper easy to follow.

- The experimental results provide strong support for the proposed method, demonstrating efficient training and comparable performance to existing approaches.

**Weaknesses:**

- The authors overlook an existing work that also focuses on improving the efficiency of data annotation in online DPO, namely “Optune: Efficient Online Preference Tuning.” Therefore, the claim that this paper presents the first data selection framework to minimize annotation costs for online DAP methods appears to be overstated. A direct comparison between DOTA and Optune would strengthen the contribution.

- The function of the proposed Preference Perplexity (PFP) metric appears to be similar to the DPO loss formulation. The paper should clarify the differences between filtering using DPO loss and PFP. In addition, regarding computational efficiency, the benefit of requiring only a single forward pass is marginal compared to the dominant cost of response generation and model updating.

- The scalability of DOTA is questionable. The clustering algorithm used is not well-scalable, which may limit its applicability to larger datasets in future studies.

**Questions:**

Good presentation, no questions.

---

### Official Review · Reviewer_kWyP · 2025-10-31

**Soundness:** 2
**Presentation:** 2
**Contribution:** 2
**Rating:** 4
**Confidence:** 3

**Summary:**

The paper introduces Preference Perplexity, a novel metric designed to identify and prioritize the most informative samples for online preference alignment. By quantifying uncertainty or diversity in preference data, this metric enables more efficient selection of training samples that better guide model alignment. Building on this concept, the authors propose the DOTA (Dynamic Online Training and Adaptation) algorithm, which effectively balances exploration (selecting uncertain or diverse samples) and exploitation (refining on high-quality known samples) at each training iteration. Experimental results across multiple downstream tasks demonstrate that this sample selection strategy significantly improves alignment efficiency and overall model performance compared to standard online RLHF or preference-based optimization baselines.

**Strengths:**

Strong performance compared to the baseline approaches, the algorithm is simple yet effective and the structure is clear and flows naturally.

**Weaknesses:**

There lacks ablation study analyzing the contribution of each component to the final results. For example, it would be helpful to see the outcome if clustering were removed entirely, if inference were performed on the reference model as well, or how much performance delta is observed when using the approximation method for sample selection.

**Questions:**

The author might have already run these, I haven't found:

* if inference were performed on the reference model as well, or how much performance delta is observed when using the approximation method for sample selection.

* If we don't do clustering, how would the performance be?

---

### Official Review · Reviewer_XQFh · 2025-11-01

**Soundness:** 2
**Presentation:** 3
**Contribution:** 2
**Rating:** 4
**Confidence:** 4

**Summary:**

Annotating preference data is crucial but expensive in online direct alignment from preference (DAP). To reduce the cost, this paper introduces a new data selection metric Preference Perplexity (PFP) and a multi-armed bandit (MAB) based selection algorithm. PFP approximates DAP losses by computing the deviation between the preference probability from the policy model in a single pass. MAB avoids pre-generating responses for all candidate questions, but dynamically samples from a set of clusters. Results show better performance with good efficiency.

**Strengths:**

* The proposal of PFP that adopts approximated DAP losses for data selection, beyond scoring gaps from reward models
* The experiments are conducted with different LLMs, different model sizes, and several benchmarks
* The proposed method performs well at different data selection ratios

**Weaknesses:**

* The method mainly works for iterative DAP methods, i.e. semi-online DAP, not real online post-training.
* The lack of efficiency improvement estimation: as iteration increases, i.e. approaching the true online learning, directly annotating preference pairs from the reward model may deliver the best quality-efficiency trade-off since selecting data offline incurs non-ignorable cost.
* The theoretical analysis in Section 3.3 adopts an unrealistic assumption, where policy model and reference model are the same or similar. In fact, as training processes, the policy model will depart from the reference model. Analysis in Section 4.3 is uninformative; a better measure is the KL divergence between these two models during training.
* The proposed PFP, as explained in Equation 9, has a clear preference over response pairs of different lengths. This may enforce the length bias issue in post-training, but no relevant discussion in the paper.

**Questions:**

1. In lines 60-62, the authors argue that “the score gap produced by a reward model … does not necessarily reflect its potential benefit to the target model.” Do you have any empirical evidence supporting this? For example, did examples selected by PFP have lower reward score gaps?
2. In line 64, “the expected preference probability produced by the reward model”, do you mean reference model?

---

### Official Review · Reviewer_LfqV · 2025-11-02

**Soundness:** 3
**Presentation:** 3
**Contribution:** 3
**Rating:** 4
**Confidence:** 3

**Summary:**

The paper tackles the problem of efficient post-training alignment and proposes DOTA, a data selection framework that minimizes the cost of generating preference data. To enable better data selection, it proposes a computationally efficient metric called Preference Perplexity (PFP), which approximates the DAP loss gradient with a single forward pass to estimate the value of a preference data point. The paper also uses a multi-armed bandit (MAB) strategy that operates on clusters of questions, allowing it to select valuable questions for annotation without needing to generate responses for the entire candidate pool.  Three models - LLaMA-3-8B, Qwen-3-4B, Qwen-3-1.7B are used to evaluate DOTA's efficiency, with 3x lower computational cost than full dataset training while maintaining or increasing performance on downstream tasks.

**Strengths:**

- LLM alignment is an expensive and open problem; the proposed method would significantly lower the required compute.
- Selecting valuable questions before the expensive response generation step—is a novel and significant departure from prior work like SeRA and Less is More
- The authors conduct extensive experiments across multiple models, datasets, and downstream evaluation benchmarks. The consistent outperformance over strong baselines in terms of both performance and computational efficiency (EFLOPs) provides strong emperical backing for DOTA.
- The papers show that the approach is generic for not only DPO but also IPO and SLiC, with positive results.

**Weaknesses:**

- The framework's performance hinges on the assumption that semantically similar questions (as determined by embeddings) will yield preference data with similar PFP scores. This does not seem sufficiently motivated.
- The ablation study (Figure 3b) shows that performance is quite sensitive to the number of clusters, k. While the "Elbow method" is mentioned for selecting k, this can be heuristic and unstable. The paper would benefit from a more detailed discussion on the robustness of this choice and practical guidance on how to set k for new datasets without extensive tuning.
- It is unclear whether the preprocessing steps are part of the efficiency calculations or how significant they are.
- While quantitative results are strong, the paper is missing a qualitative analysis of the data being selected. What do the high-PFP questions/responses actually look like? A small-scale human study would also strengthen the paper and reduce concerns of any reward hacking.

**Questions:**

- Could you provide more justification for the core assumption that semantic clustering of questions effectively groups them by their potential PFP scores?
- Could you comment on the one-time computational cost of the initial question embedding and clustering step?
- How does the performance of DOTA vary with the quality of the reward model used for generating preference labels?
- Could you provide a few qualitative examples of the prompts and response pairs that receive the highest PFP scores? This would provide valuable intuition as to whether the metric is identifying genuinely challenging examples or potentially exploiting reward model artifacts.

---

### Note · Authors · 2025-12-30

I have read and agree with the venue's withdrawal policy on behalf of myself and my co-authors.